# NMR-Based Metabolomic Analysis on the Protective Effects of Apolipoprotein A-I Mimetic Peptide against Contrast Media-Induced Endothelial Dysfunction

**DOI:** 10.3390/molecules26175123

**Published:** 2021-08-24

**Authors:** Ting Jiang, Qian Du, Caihua Huang, Wenqi Xu, Ping Guo, Wei Li, Xianwei Xie, Yansong Guo, Donghui Liu, Donghai Lin

**Affiliations:** 1Key Laboratory for Chemical Biology of Fujian Province, MOE Key Laboratory of Spectrochemical Analysis & Instrumentation, College of Chemistry and Chemical Engineering, Xiamen University, Xiamen 361000, China; xmujiangt@126.com (T.J.); piaolongqi@163.com (W.X.); 2Department of Cardiology, Guangzhou Red Cross Hospital, Medical College, Jinan University, Guangzhou 510000, China; duqian1213@126.com; 3Research and Communication Center of Exercise and Health, Xiamen University of Technology, Xiamen 361000, China; huangcaihua@xmut.edu.cn; 4Department of Cardiology, The Affiliated Cardiovascular Hospital of Xiamen University, Medical College of Xiamen University, Xiamen 361000, China; pingguolulu420@163.com (P.G.); lwcardiology@126.com (W.L.); 5Department of Cardiology, Fujian Provincial Hospital, Fujian Cardiovascular Institute, Fujian Provincial Key Laboratory of Cardiovascular Disease, Provincial Clinical Medicine College of Fujian Medical University, Fuzhou 350000, China; xianwei1123@163.com (X.X.); ysguo1234@163.com (Y.G.); 6Department of Geriatrics, Guangzhou First People’s Hospital, School of Medicine, South China University of Technology, Guangzhou 510000, China

**Keywords:** apoA-I mimetic peptide, contrast media, endothelial cell, metabolomics, NMR

## Abstract

Endothelial dysfunction plays key roles in the pathological process of contrast media (CM)-induced acute kidney injury (CI-AKI) in patients undergoing vascular angiography or intervention treatment. Previously, we have demonstrated that an apolipoprotein A-I (apoA-I) mimetic peptide, D-4F, inhibits oxidative stress and improves endothelial dysfunction caused by CM through the AMPK/PKC pathway. However, it is unclear whether CM induce metabolic impairments in endothelial cells and whether D-4F ameliorates these metabolic impairments. In this work, we evaluated vitalities of human umbilical vein endothelial cells (HUVECs) treated with iodixanol and D-4F and performed nuclear magnetic resonance (NMR)-based metabolomic analysis to assess iodixanol-induced metabolic impairments in HUVECs, and to address the metabolic mechanisms underlying the protective effects of D-4F for ameliorating these metabolic impairments. Our results showed that iodixanol treatment distinctly impaired the vitality of HUVECs, and greatly disordered the metabolic pathways related to energy production and oxidative stress. Iodixanol activated glucose metabolism and the TCA cycle but inhibited choline metabolism and glutathione metabolism. Significantly, D-4F pretreatment could improve the iodixanol-impaired vitality of HUVECs and ameliorate the iodixanol-induced impairments in several metabolic pathways including glycolysis, TCA cycle and choline metabolism in HUVECs. Moreover, D-4F upregulated the glutathione level and hence enhanced antioxidative capacity and increased the levels of tyrosine and nicotinamide adenine dinucleotide in HUVECs. These results provided the mechanistic understanding of CM-induced endothelial impairments and the protective effects of D-4F for improving endothelial cell dysfunction. This work is beneficial to further exploring D-4F as a potential pharmacological agent for preventing CM-induced endothelial impairment and acute kidney injury.

## 1. Introduction

As the predominant diagnostic reagents, contrast media (CM) have been extensively used in clinical angiography and intravascular catheter-based intervention [1]. Previous works have demonstrated the adverse effects of CM for patients, in particular for those high-risk patients with diabetes mellitus, heart failure and chronic kidney disease [2]. Water-soluble iodinated CM can be excreted through the kidneys, and CM-induced acute kidney injury (CI-AKI) has become the third leading cause of new-onset renal failure in hospitalized patients with CM injection [3]. The pathophysiological mechanisms underlying CI-AKI primarily include oxidative stress, inflammation, endothelial dysfunction, and renal tubular toxicity [2]. More importantly, CM are given via circulation during the diagnostic and interventional processes, which usually impair vascular endothelium and subsequently contribute to systemic and organ-specific adverse reactions.

As an iso-osmolar non-ionic iodinated CM, iodixanol (also named as visipaque) has a high contrast effect with fewer side effects than hypertonic and ionic CM such as meglumin diatrizoate [4]. In our previous work, we revealed that iodixanol impairs cell viability, promotes vascular cell adhesion molecule-1 (VCAM-1) and intercellular cell adhesion molecule-1 (ICAM-1) expression, and induces cell apoptosis in human umbilical vein endothelial cells (HUVECs) [5]. Furthermore, we indicated that iodixanol induces the phosphorylation of protein kinase C (PKC) beta II, p47, Rac1, and endothelial nitric oxide synthase at Thr495, eliciting ROS release and ONOO^−^ generation [5].

HUVECs regulate physiological functions of blood vessels, such as vascular tone, blood coagulation and thrombosis [6]. Endothelial dysfunction is usually related to the imbalance of many factors, including a reduction in nitric oxide (NO) and production of endothelin-1 (ET-1) and thromboxane A2 (TXA2), which exhibit the features of pro-inflammation, pro-oxidation, pro-coagulation and pro-vascular adhesion [7]. In patients with CM injection, the endothelium of micro-vessels and macro-vessels are exposed to high doses of circulating CM. These chemical reagents may trigger endothelial dysfunction and contribute to systemic and kidney-specific adverse reactions, which subsequently cause CI-AKI initiation and progression [8]. CM inhibit the release of NO in endothelial cells in vitro and suppress endothelium-dependent arterial dilation in diabetic patients undergoing vascular angiography and intervention in vivo [9,10]. Moreover, CM increase ROS generation, vascular cell adhesion molecule-1 expression, and inflammatory factors (e.g., TNF-alpha and IL-6) release in endothelial cells [11,12]. In addition, CM also decrease endothelial cell viability and cause apoptosis and necrosis, reducing blood flow velocity in renal capillaries and contributing to renal medullary hypoxia and tubular failure [1,13]. However, how CM induce metabolic impairments in endothelial cells remains to be clarified.

Previous studies have reported that both high-density lipoprotein (HDL) and apolipoprotein A-I (apoA-I) have significant protective effects on endothelial cells [14]. An apoA-I mimetic peptide, D-4F, shares similar functions to native apoA-I and also displays remarkable protection of endothelial cells [15]. D-4F can promote endothelial nitric-oxide synthase (eNOS) activation and NO production in coronary artery endothelial cells, protecting against myocardial ischemia/reperfusion impairments in mice [16]. Furthermore, we also found that D-4F can scavenge intracellular ROS via upregulating heme oxygenase-1 (HO-1) expression and decreasing ROS production through inhibiting NADPH oxidase activation and eNOS uncoupling, playing anti-oxidative, anti-inflammatory and anti-apoptotic roles in endothelial cells [5,17]. Even if iodixanol induces oxidative stress through NOX activation and ONOO^−^ formation, D-4F can significantly inhibit NOX activation and eNOS dysregulation via the AMPK/PKC pathway and decrease ROS production and ONOO^−^ formation, thus ameliorating apoptosis and inflammation induced by iodixanol [5,17]. These results indicate the significant anti-oxidative effects of D-4F in endothelial cells through scavenging ROS and suppressing ROS generation. However, it is unclear whether D-4F could alleviate the CM-induced metabolic impairments in endothelial cells.

As is known, biological functions of endothelial cells are driven by cellular metabolism. Expectedly, endothelial dysfunction is closely related to metabolic impairments in endothelial cells. Previous studies have demonstrated that metabolomic analysis can help to elucidate overall abnormal changes in endothelial cells in pathological conditions [18,19]. We previously indicated that D-4F can alleviate oxidized-low density lipoprotein (ox-LDL)-induced oxidative stress and abnormal glycolysis in endothelial cells by performing NMR-based metabolic profiling [20]. In this work, we conducted NMR-based metabolomic analysis to address CM-induced metabolic impairments and exploit the metabolic mechanisms underlying the protective effects of D-4F in endothelial cells. Our results showed that while CM treatment significantly impaired several metabolic pathways, including glucose metabolism, glutathione biosynthesis, TCA cycle and choline metabolism, D-4F pretreatment could effectively alleviate CM-induced metabolic impairments. Our work may be of benefit to the exploitation of D-4F as a potential pharmacological agent against CM-induced endothelial impairments and CI-AKI.

## 2. Results

### 2.1. Protective Effects of D-4F on HUVEC Vitality Impaired by Iodixanol

To explore the impaired effects of iodixanol and the protection of D-4F on the vitality of HUVECs, we incubated HUVECs with different doses of iodixanol and D-4F. The numbers of HUVECs were significantly reduced with iodixanol treatment in dose- and time- dependent manners (Figure 1A–C). These results indicated that iodixanol significantly impaired the vitality of HUVECs, and D-4F pretreatment distinctly increased cell numbers and greatly improved the iodixanol-impaired vitality of HUVECs (Figure 1D).

### 2.2. ^1^H NMR Spectra of Intracellular Metabolites

We recorded 1D ^1^H-NMR spectra on aqueous extracts derived from iodixanol-treated HUVECs with or without D-4F pretreatment (Figure 2). In the spectra, peaks 1–4 represent the four resonances of iodixanol, which are overlapped with the resonances of four metabolites including acetate, succinate, glycine and ethylene glycol. In total, 36 metabolites were identified based on the NMR spectra. Spectra of 2D ^1^H-^13^C HSQC and ^1^H-^1^H TOCSY were used to confirm the resonance assignments of the metabolites (Appendix A). The resonance assignments are displayed in Appendix A. As expected, iodixanol treatment reduced intracellular levels of AXP (overlapped ATP/ADP/AMP) in HUVECs (Figure 2B). Significantly, D-4F pretreatment partially restored the levels of AXP that were reduced by iodixanol (Figure 2C,D).

### 2.3. Metabolic Profiles of HUVECs

We performed PCA to exploit metabolic profiles of the five groups of HUVECs (Ctrl, Iod10, Iod30, D-4F + Iod10 and D-4F + Iod30). The score plots of the PCA models show that the two iodixanol-treated groups were metabolically distinct from the Ctrl group, and the metabolic discrimination was iodixanol dose-dependent (Figure 3A). Similarly, the metabolic profiles of the D-4F + Iod10 and D-4F + Iod30 groups were distinctly different from those of the Ctrl group, and the D-4F + Iod30 cells were more clearly distinguished from the controls compared to D-4F + Iod10 cells (Figure 3B,C).

### 2.4. Significant Metabolites Identified in HUVECs

We performed the supervised OPLS-DA analysis to identify the significant metabolites primarily contributing to the metabolic distinctions among the five groups of HUVECs. The score plots of the OPLS-DA models display that the Iod10 and Iod30 groups were obviously discriminated from the Ctrl group (Figure 4A,B), and the D-4F + Iod10 and D-4F + Iod30 groups were remarkably distinguished from the Iod10 and Iod30 groups (Figure 4C,D). The robustness of the established OPLS-DA models was assessed by using response permutation tests with 200 cycles (Appendix A). The values of R2Y and Q2 reflect the reliabilities of the models, where R2Y represents the explanation and Q2 represents the accuracy of prediction. The closer the values of R2Y and Q2 are to 1, the more reliable the OPLS-DA model is.

The loading plots of the OPLS-DA models were used to identify the significant metabolites for the five groups with VIP ≥ 1 and the correlation coefficient |r| ≥ the critical value corresponding to statistical significance *p* < 0.05 (Figure 4). In total, 16 significant metabolites were identified from Iod10 vs. Ctrl (Figure 4A, 2 increased and 14 decreased metabolites), and 23 metabolites were identified from Iod30 vs. Ctrl (Figure 4B, 1 increased and 22 decreased metabolites). Compared with the Ctrl group, the Iod30 group shared 8 decreased metabolites with the Iod10 group including pantothenate, leucine, isoleucine, lactate, threonine, tyrosine, phenylalanine and formate. Moreover, 17 significant metabolites were identified from D-4F + Iod10 vs. Iod10 (Figure 4C, 5 increased and 12 decreased metabolites) and 22 significant metabolites were identified from D-4F + Iod30 vs. Iod30 (Figure 4D, 22 increased metabolites). The D-4F + Iod30 group shared four increased metabolites with the D-4F + Iod10 group including ethanol, glutamate, glutamine, and NAD^+^. The D-4F + Iod30 group contained 13 significant metabolites related to amino acid metabolism (valine, leucine, isoleucine, threonine, alanine, lysine, proline, glutathione, aspartate, creatine, taurine, tyrosine, and phenylalanine), and two significant metabolites associated with glucose metabolism (lactate, and succinate).

### 2.5. Differential Metabolites Identified in HUVECs

To quantitatively compare metabolite levels among the five groups of HUVECs, we calculated the absolute metabolite levels (mM) based on their peak integrals relative to the peak integral of TSP as an inner reference molecule with a known concentration (Table 1). Four metabolites (acetate, succinate, glycine, and ethylene glycol) were excluded from the calculation as their resonances were overlapped with iodixanol. Table 1 shows the fold change values of the absolute metabolite levels of Iod10/Ctrl, D-4F + Iod10/Iod10, Iod30/Ctrl, D-4F + Iod30/Iod30.

We then performed Student’s *t* tests to compare the pair-wise absolute metabolite levels between the groups (Appendix A). Differential metabolites were identified with a criterion of *p* < 0.05. In total, 12 differential metabolites were identified from Iod10 vs. Ctrl, and 23 different metabolites were identified from Iod30 vs. Ctrl (Table 1). Moreover, 17 different metabolites were identified from D-4F + Iod10 vs. Iod10, and 16 different metabolites were identified from D-4F + Iod30 vs. Iod30 (Table 1). The fold change values of metabolites were also shown (Appendix A).

### 2.6. Characteristic Metabolites Identified in HUVECs

We identified the characteristic metabolites between the five groups with the two criteria of VIP ≥ 1 calculated from the OPS-DA models and *p* < 0.05 calculated from the pair-wise Student’s *t* tests. We identified seven characteristic metabolites from the comparison of Iod10 vs. Ctrl, including threonine, sn-glycero-3-phosphocholine (GPC), leucine, creatine, glucose, glutamine, and NAD^+^ (Figure 4A, four increased and three decreased metabolites). We also identified 18 characteristic metabolites from the comparison of Iod30 vs. Ctrl, including three increased and 15 decreased metabolites (Figure 5A). Furthermore, we identified 15 characteristic metabolites from the comparison of D-4F + Iod10 vs. Iod10, including four increased and eleven decreased metabolites, and 15 characteristic metabolites from that of D-4F + Iod30 vs. Iod30, including eleven increased and four decreased metabolites (Figure 5B).

The comparison of Iod10 vs. Ctrl shared six characteristic metabolites with that of Iod30 vs. Ctrl, including glutamine, threonine, GPC, leucine, creatine, and NAD^+^ (Figure 5A). Moreover, the comparison of D-4F + Iod10 vs. Iod10 shared 7 characteristic metabolites with that of D-4F + Iod30 vs. Iod30, including glutamine, glutamate, O-phosphocholine (PC), GPC, succinate, tyrosine, and NAD^+^ (Figure 5B). Three characteristic metabolites (glutamine, GPC and NAD^+^) were shared by the four pair-wise comparisons, i.e., Iod10 vs. Ctrl, Iod30 vs. Ctrl, D-4F + Iod10 vs. Iod10, and D-4F + Iod30 vs. Iod30.

### 2.7. Significant Altered Metabolic Pathways Identified in HUVECs

To mechanistically understand the effects of either iodixanol treatment or D-4F pretreatment on the metabolic profiles of HUVECs, we identified the significantly altered metabolic pathways (significant pathways) by performing the metabolic pathway analyses of Iod10 vs. Ctrl, Iod30 vs. Ctrl, D-4F + Iod10 vs. Iod10, and D-4F + Iod30 vs. Iod30. In total, 10 significant pathways were identified, including (a) glutamine and glutamate metabolism; (b) alanine, aspartate and glutamate metabolism; (c) glycine, serine and threonine metabolism; (d) phenylalanine metabolism; (e) phenylalanine, tyrosine and tryptophan biosynthesis; (f) nicotinate and nicotinamide metabolism; (g) starch and sucrose metabolism; (h) glutathione metabolism; and (i) taurine and hypotaurine metabolism; (j) Histidine metabolism (Figure 6). 

The two comparisons of Iod10 vs. Ctrl and Iod30 vs. Ctrl shared seven significant pathways (a, b, c, d, e, f, h) (Figure 6A,B). The Iod10 group had one extra significant pathway (g), while the Iod30 group had another one (i). Furthermore, the comparison of D-4F + Iod10 vs. Iod10 identified nine significant pathways (a, b, c, d, e, f, g, h, j) (Figure 6C), sharing eight pathways with the comparison of Iod10 vs. Ctrl (Figure 5A) with one extra pathway (j). Additionally, the comparison of D-4F + Iod30 vs. Iod30 identified four significant pathways (a, b, i, j) (Figure 6D), sharing two pathways (a, b) with the comparison of Iod30 vs. Ctrl (Figure 6B). Interestingly, the taurine and hypotaurine metabolism (i) was only identified in the two comparisons of Iod30 vs. Ctrl and D-4F + Iod30 vs. Iod30, while the histidine metabolism (j) only appeared in the two comparisons of D-4F + Iod10 vs. Iod10 and D-4F + Iod30 vs. Iod30. It seemed that the pathway (i) was related to Iod30 treatment, and the pathway (j) was associated with D-4F pretreatment.

We depicted the significant pathways identified from the pair-wise comparisons of Iod10 vs. Ctrl and Iod30 vs. Ctrl based on the KEGG database (Appendix A). It was suggested that iodixanol treatment primarily impaired five metabolic pathways: glutathione metabolism; glycerophospholipid metabolism; alanine, aspartate, and glutamate metabolism; glucose metabolism; glutamine and glutamate metabolism. 

We also depicted the significant pathways identified from the comparison of D-4F + Iod10 vs. Iod10 (Appendix A). Significantly, D-4F pretreatment also altered the five metabolic pathways impaired by iodixanol, potentially playing a crucial role in antagonizing oxidative stress via glucose metabolism. Moreover, we depicted the significant pathways identified from the comparison of D-4F + Iod30 vs. Iod30 (Appendix A), including glutathione metabolism; glycerophospholipid metabolism; alanine, aspartate, and glutamate metabolism; glutamine and glutamate metabolism; taurine and hypotaurine metabolism. The first four significant pathways were shared by the comparison of D-4F + Iod10 vs. Iod10. It was noted that the taurine and hypotaurine metabolism was significantly altered only in the comparison of D-4F + Iod30 vs. Iod30, implying that D-4F pretreatment could alleviate oxidative stress induced by iodixanol treatment at 30 mg I/mL.

## 3. Discussion

With the increasing use of CM, CI-AKI becomes the third most common cause of hospital-acquired renal failure [3,21]. Endothelial dysfunction plays important roles in the pathogenesis of CM-induced clinical complications [2,8]. We have previously demonstrated that D-4F can scavenge excessively generated ROS via up-regulating HO-1 and decrease ROS production through inhibiting NOX activation and eNOS uncoupling, exerting anti-oxidative, anti-inflammatory and anti-apoptotic effects in endothelial cells [5,17]. Furthermore, we performed an NMR-based metabolomic analysis to mechanistically understand how D-4F alleviates ox-LDL-induced oxidative stress and disordered glycolysis in endothelial cells [20]. Here, we wondered whether D-4F could protect endothelial cells from CM-induced metabolic disorders and which mechanisms underly the protective effects. To address these issues, we performed metabolic profiling to understand the molecular mechanisms underlying the protective effects of D-4F on endothelial cells against iodixanol. We observed that iodixanol treatment impaired the vitality of HUVECs, significantly impaired the metabolic profiles of HUVECs, and D-4F pretreatment could greatly improve the iodixanol-impaired cell vitality and profoundly ameliorate iodixanol-induced metabolic impairments in HUVECs.

### 3.1. D-4F Alleviates Iodixanol-Induced Intracellular Oxidative Stress

Iodixanol induces disorders in five metabolic pathways related to oxidative stress, including glutamine and glutamate metabolism; alanine, aspartate and glutamate metabolism; glycine, serine and threonine metabolism; glutathione metabolism; taurine and hypotaurine metabolism. Previous work reported that prolonged and severe oxidative stress was observed in hemodialysis patients with CM injections [22]. In this study, we found that several antioxidative molecules (glutathione, creatine, and fumarate) were decreased in iodixanol-treated HUVECs. These results imply that the antioxidative defensive system was impaired in endothelial cells. As is known, oxidative stress plays an important role in the cytotoxicity of CM [23]. Our results also showed iodixanol-induced metabolic disorders and oxidative stress in HUVECs, as indicated by the inhibited glutathione metabolism and obviously decreased levels of glutamate (the precursor for glutathione synthesis) in iodixanol-treated HUVECs. 

As cysteine was not assigned in the 1D ^1^H-NMR spectra of aqueous extracts derived from HUVECs, we had not measured intracellular levels of cysteine (the third precursor for glutathione synthesis). Nevertheless, we observed decreased levels of methionine (the precursor of cysteine) in HUVECs after Iod30 treatment, indicating that iodixanol inhibited glutathione biosynthesis. Some studies showed that CM enhance intracellular ROS release in a dose-dependent manner [11,23], and glutathione can directly scavenge ROS and indirectly act as a cofactor of specific antioxidative enzymes for protecting cells against oxidative stress [24]. Expectedly, reduced glutathione could not fully scavenge ROS. We also observed that a variety of glucogenic amino acids (alanine, aspartate, glutamate, proline, tyrosine, phenylalanine, leucine, isoleucine, and valine) were largely consumed in the Iod30 group. These amino acids could participate in TCA cycle and activate oxidative phosphorylation, as indicated by up-regulated succinate levels in iodixanol-treated cells. As oxidative phosphorylation is the major source of intracellular ROS production [25], the activated oxidative phosphorylation would produce more ROS, further aggravating intracellular oxidative stress.

On the other hand, the comparison of D-4F + Iod10 vs. Iod10 shared seven characteristic metabolites with that of Iod10 vs. Ctrl, and we hence speculated that D-4F might play a protective role by reversing the levels of these metabolites. We found that the glutathione metabolism was promoted in the D-4F group. Although the glutathione level was not changed obviously, the glutamate level increased in HUVECs pretreated with D-4F. Glutaminase is highly expressed in endothelial cells, which catabolizes glutamine to glutamate, and inhibition of glutaminase is associated with cell aging and proliferation [26]. Therefore, the observation that D-4F restored the glutamate level decreased by iodixanol, might indicate the activation of glutaminase. Note that the activity of glutaminase needs to be exploited in the future. Furthermore, we found that iodixanol-induced increase in succinate was reversed by D-4F. These observations suggest that TCA cycle is potentially decelerated in response to D-4F pretreatment. Therefore, D-4F might alleviate oxidative stress by increasing glutathione biosynthesis in the D-4F + Iod10 group. 

Besides, glutathione biosynthesis also contributed to the inhibition of glycolysis in the D-4F + Iod10 group. Previous studies have demonstrated that NO plays a protective role in kidney injuries by slowing the glycolytic pathway and increasing the pentose phosphate pathway (PPP) [27]. We found that the levels of glucose and lactate decreased in the D-4F + Iod10 group, indicating that D-4F inhibited glycolysis in HUVECs. We supposed that glucose might participate in PPP, producing pentose and NADPH [28]. NADPH increases reduced glutathione (GSH) levels and activates antioxidative enzymes, which can restore the function of proteins damaged by oxidative stress [29]. Future work is required to confirm the enhanced PPP by using the isotope labeling technique. These results indicate that D-4F pretreatment alleviates oxidative stress induced by Iod10 treatment through inhibiting glycolysis and increasing glutathione biosynthesis. 

In the D-4F + Iod30 group, although the taurine level decreased, the taurine and hypotaurine metabolism was slightly promoted. Besides the increased levels of glutathione, creatine and fumarate, the levels of five glucogenic amino acids (glutamate, proline, threonine, tyrosine, and phenylalanine) also showed restoring tendencies in response to D-4F. These results indicate that D-4F improves oxidative stress by promoting glutathione metabolism and reducing the consumption of amino acids which are related to the TCA replenishment pathway.

### 3.2. D-4F Restores the Energy Production Impaired by Iodixanol

Previous work showed that accumulation of CM in the kidneys enhances local inflammatory mediators and ROS release, leading to cellular energy production disorders and damaging local tissues [30]. Increased ROS in endothelial cells under hypoxia produces hypoxia-inducible factor-1α (HIF-1α) and activates intracellular glycolysis [31]. In this study, increased lactate was detected in HUVECs treated by iodixanol, indicative of an enhanced glycolytic process. Furthermore, the NMR spectra show decreased peak heights of AXP in the Iod10 and Iod30 groups, indicative of decreased levels of AXP. These results were associated with iodixanol-induced oxidative stress. Despite iodixanol activated glycolysis, the energy production in HUVECs did not significantly increase, indicating that the energy production from glycolysis was blocked. We found that the lactate level was reduced in the D-4F + Iod10 group. Furthermore, the metabolic pathway analysis indicated that the glycine, serine and threonine metabolism was also down-regulated in the D-4F + Iod10 group. As serine is a positive activator of pyruvate kinase (PKM2) [32], the decreased level of serine would contribute to reduced glycolysis [33].

As shown in Figure 2B, resonances of AXP (overlapped ATP/ADP) exhibited significant peak height differences among the three groups (Ctrl, Iod10, Iod30), reflecting that the iodixanol treatment reduced intracellular levels of AXP in HUVECs. Significantly, the D-4F pretreatment partially restored the peak heights of AXP that were reduced by the iodixanol treatment (Figure 2C,D).

D-4F pretreatment partially restored the NMR peak heights of AXP in the D-4F + Iod10 group relative to the Iod10 group, and the D-4F + Iod30 group relative to the Iod30 group. As it is known, energy production from Warburg effect could facilitate cell proliferation [34]. The reduction of energy production decelerates the proliferation and migration of endothelial cells, thereby inhibiting angiogenesis [35]. Previous works have demonstrated that CM can not only inhibit the proliferation and migration, but also promote apoptosis in endothelial cells [9,36], while D-4F can promote endothelial cell proliferation though the eNOS/NO pathway [37]. Note that NO can inhibit the glycolytic metabolism [27]. Therefore, it could be supposed that D-4F might promote HUVEC proliferation by inhibiting glycolysis and enhancing AXP production, thereby improving cell viability. 

Generally, cellular energy status could be reflected by the levels of ATP, ADP and NAD^+^ acting as a critical factor for cell survival [38,39]. NAD^+^ usually appears as an electron acceptor in glycolysis and the Krebs cycle [40], and is de novo synthesized from tryptophan and supplanted from nicotinamide (NAM) [41]. Although tryptophan (the precursor of NAD^+^ biosynthesis) was not assigned in the recorded NMR spectra of HUVECs, tyrosine (the precursor of tryptophan) was detected. We found that the tyrosine level decreased in the Iod30 group but increased in the D-4F + Iod30 group. Previous studies have indicated that acute kidney injury (AKI) patients are characterized by the disorder of de novo biosynthesis of NAD^+^ [42]. Our study showed that D-4F improves de novo biosynthesis of NAD^+^ that can also prolong the life span of age-associated complications including AKI [43,44]. Furthermore, it was reported that the levels of NAD^+^ and NADP^+^ decreased in the kidney tissues of mice with ischemic AKI [42]. Significantly, the characteristic metabolite NAD^+^, was down-regulated in the Iod10 and Iod30 groups, but notably up-regulated in the D-4F + Iod10 and D-4F + Iod30 groups. Previously, the decreased NAD^+^ level was also reported in diseases associated with oxidative stress [45]. Our results suggest that D-4F might alleviate oxidative stress through increasing de novel biosynthesis of NAD^+^, and the up-regulated levels of ATP, ADP and NAD^+^ would be beneficial to restoring cell viability.

### 3.3. D-4F Enhances Iodixanol-Inhibited Choline Metabolism

Previous works indicated that almost all CM exclusively induce cell apoptosis and destroy the integrity of endothelial cell membrane [46], and choline plays an important role in cell division and proliferation [47]. Our work showed that iodixanol treatment decreased PC levels and increased GPC levels, while D-4F pretreatment contrarily increased PC levels and decreased GPC levels, indicating that D-4F could promote the catabolism of glycerophosphate choline and partially restore the intracellular phospholipid metabolism impaired by iodixanol. Therefore, the suppression of choline metabolism is potentially associated with the inhibition of HUVEC proliferation.

## 4. Materials and Methods

### 4.1. Cell Culture

Human umbilical vein endothelial cells (HUVECs) were isolated and cultured according to previous methods [48]. This study protocol was approved by the Ethics Committee of Fujian Provincial Hospital. HUVECs at 3–5 generations were seeded in a 10 cm Petri dish at a density of 1 × 10^6^ cells/dish. Cells were grown at 37 °C with 5% CO_2_ and cultured with endothelial cell medium (ECM; ScienCell, Carlsbad, CA, USA) containing 10% fetal bovine serum (FBS; ScienCell, Carlsbad, CA, USA) and 1% endothelial cell growth supplement (ECGS; ScienCell, Carlsbad, CA, USA). When the cell density reached 85–90%, cells were treated with different doses of iodixanol (Visipaque, 320 mg I/mL, GE Healthcare Company, San Ramon, CA, USA). Before treatment, HUVECs were deprived of serum with 0.5% FBS-ECM for 6 h. The control group (Ctrl) was treated with PBS (HyClone, Logan, Utah, USA). HUVECs were pretreated with or without D-4F (20 μg/mL) for 8 h, and then incubated with 10 or 30 mg I/mL of iodixanol for 12 h. Cell pictures were taken with a microscope (Olympus, Tokyo, Japan) in high-power fields (100×) and cell numbers were counted using Image J software.

### 4.2. Cell Metabolite Extraction

Around 5 × 10^6^ cells were harvested in each dish. The intracellular metabolites were extracted following the two-liquid extraction approach previously described [49]. Cells were washed three times with pre-cooled PBS (pH 7.4), and then 3 mL of quencher (−20 °C pre-cooled methanol) was added to each Petri dish, scraping off cells into a new centrifuge tube. Subsequently, HPLC grade chloroform and ultrapure H_2_O were added into the centrifuge tube (volume ration = 1:1:0.95). At the end, the upper aqueous phase was blown with nitrogen and lyophilized into powder.

### 4.3. NMR Sample Preparation and NMR Spectrum Acquisition

The lyophilized samples were dissolved with 18.3 μL of PBS (1.5 M K_2_HPO_4_/NaH_2_PO_4_, pH 7.4) and 531.7 μL of D_2_O, containing 1 mM sodium 3-(trimethylsilyl)-propionate-2,2,3,3-d4 (TSP), vortexed and centrifuged (4 °C, 12,000 rpm, 5 min) to remove precipitates. Finally, the collected supernatant (about 500 μL per sample) was transferred to 5 mm NMR tube. 1D ^1^H-NMR spectra were recorded on a Bruker AVANCE III HD 850 MHz (Bruker Bio Spin, Germany) at 25 °C using a TCI cryoprobe. The pulse sequence NOESYGPPR1D [(RD)-90°-t_1_-90°-τ_m_-90°-ACQ] with water suppression was used to record the NMR spectra. The relaxation delay time (RD) was 2 s, and the mixing time (τ_m_) was 10 ms. The spectral width was 20 ppm, and a total of 32 transients were collected into 64 K data points with an acquisition time (ACQ) of 1.88 s. The free induction delay (FID) signal was processed by a window function with a line broadening of 0.3 Hz, followed by Fourier transformation to obtain 1D ^1^H spectra. Both two-dimensional (2D) ^1^H-^13^C heteronuclear single quantum coherence (HSQC) and 2D ^1^H-^1^H total correlation spectroscopy (TOCSY) spectra were recorded to assist in resonance assignments of metabolites.

### 4.4. NMR Data Preprocessing

The original NMR spectra were processed with the MestReNova software (Version 9.0, Mestrelab Research S.L., Santiago de Compostela, Spain), including calibration, phase correction, baseline adjustment, and peak alignment. Regions from −0.1 ppm to 9.5 ppm were binned at δ 0.002 intervals to calculate the integral values. Non-overlapping peaks were selected to calculate spectral integrals of the metabolites. To eliminate the suppression effect of the water peak on the peak integration, the peak of δ 5.5–4.8 was deleted. Additionally, as the peak of iodixanol interfered with the metabolic profile analysis of HUVECs, integral values of iodixanol (δ 1.98–1.88, δ 2.47–2.40, δ 3.84–3.43, and δ 4.12–4.01) were set to zero. Resonances of aqueous metabolites were assigned by a combination of the Chenomx NMR Suite software (Version 8.4, Chenomx, Edmonton, AB, Canada), Human Metabolome Data Base (HMDB, http://www.hmdb.ca/ (accessed on 13 February 2021) and relevant literature [50]. The resonance assignments were confirmed using the 2D NMR spectra. 

### 4.5. Multivariate Analysis

The SIMCA-P software (Version 14.0, Umetrics AB, Umeå, Sweden) was used to perform multivariate analysis to differentiate metabolic profiles among the groups of HUVECs. The Pareto scaling method was used on the normalized spectral integrals to increase the magnitude of low-level metabolites without significantly extending the noise. An unsupervised principal component analysis (PCA) was conducted to reveal the trends of the metabolic profile classifications, highlight the outliers, and show the metabolic clusters among the samples. Furthermore, a supervised orthogonal projection on latent structure with discriminant analysis (OPLS-DA) was performed to further improve the metabolic classifications of the different groups and identify the potential variables significantly responsible for the metabolic distinctions. The reliabilities of established OPLS-DA models were then evaluated using response permutation tests with 200 cycles. R2 and Q2 reflect the reliabilities of the OPLS-DA models, where R2 represents the explanation and Q2 represents the accuracy of prediction. The closer the values of R2 and Q2 are to 1, the more reliable of the OPLS-DA model. 

The variable importance in projection (VIP) scores of the metabolites were calculated from the OPLS-DA models with SIMCA-P + 14.0. The OPLS-DA models were used to identify the significant metabolites primarily contributing to the metabolic separations between these groups. The significant metabolites were identified using two criteria: VIP ≥ 1; the correlation coefficient |r| ≥ the critical value corresponding to statistical significance *p* < 0.05. 

### 4.6. Univariate Analysis

The pair-wise Student’s *t*-test was conducted to quantitatively compare the absolute metabolite levels with the SPSS software (Version 22.0, Chicago, IL, USA). Metabolites with statistical significance *p* < 0.05 were identified to be differential metabolites. Characteristic metabolites were determined by two criteria: *p*< 0.05 calculated from the univariate analysis; VIP ≥ 1 calculated from the PLS-DA models.

### 4.7. Metabolic Pathway Analysis

Metabolic pathway analysis was conducted to identify significantly altered metabolic pathways induced by iodixanol and D-4F based on the levels of the assigned metabolites using the Pathway Analysis module provided by Metaboanalyst 4.0 webserver. The following parameters were selected: Enrichment method, Hypergeometric Test; Topology analysis, Relative-betweeness Centrality; Pathway library, Homo sapiens (KEGG). Metabolic pathways with −lg(*p*) scores > 3 and pathway impact values (PIVs) > 0.2 were identified to be significantly altered metabolic pathways. 

## 5. Conclusions

We have demonstrated that iodixanol treatment can profoundly decrease the endothelial cell vitality of HUVECs and significantly impair metabolic pathways associated with energy production and oxidative stress in HUVECs. Iodixanol can activate glucose metabolism and TCA cycle and inhibit choline metabolism and glutathione metabolism. D-4F pretreatment can partially improve the iodixanol-impaired vitality of HUVECs, and profoundly ameliorate iodixanol-induced metabolic impairments through improving glycolysis, TCA cycle, choline metabolism, glutathione metabolism, and glycerophospholipid metabolism. These results shed light on the molecular mechanisms underlying the impairing effects of iodixanol and protective effects of D-4F on endothelial cells. This work is of benefit to further exploration of D-4F as novel reagents to improve CM-induced endothelial dysfunction and prevent CI-AKI.

## Figures and Tables

**Figure 1 molecules-26-05123-f001:**
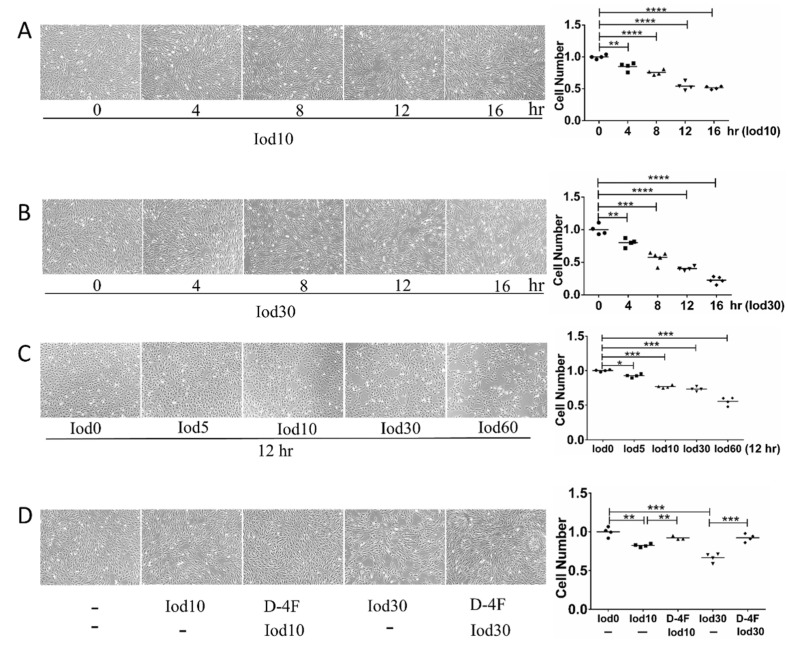
Cell vitality assay of HUVECs treated with iodixanol and D-4F. HUVECs were incubated with 10 or 30 mg I/mL of iodixanol for different time intervals (**A**,**B**), or with different doses of iodixanol for 12 h (**C**). Alternatively, HUVECs were pre-treated with or without 20 μg/mL of D-4F for 8 h and then incubated with 10 or 30 mg I/mL for 12 h (**D**). Cell images were taken at 100× magnitude. (* *p* < 0.05; ** *p* < 0.01; *** *p* < 0.001; **** *p* < 0.0001. Circles represented group Iod0; boxes represented group Iod10; upright triangle represented group D-4F + Iod10; inverted triangle represented group Iod30; diamond represented group D-4F + 30).

**Figure 2 molecules-26-05123-f002:**
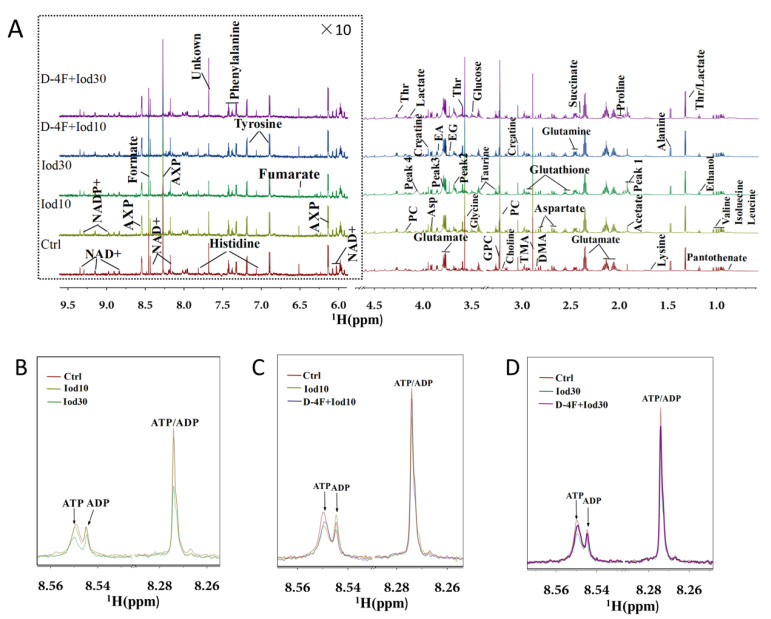
Average 850 MHz ^1^H-NMR spectra recorded on aqueous extracts of HUVECs. Full ^1^H spectra of the control group (Ctrl), Iod10-treated group (Iod10), Iod30-treated group (Iod30), D-4F with Iod10-treated group (D-4F + Iod10), and D-4F with Iod30-treated group (D-4F + Iod30) are shown (**A**). The vertical scales were kept constant in all spectra. The water region (4.8–5.5 ppm) was removed. The region of 6.0–9.5 ppm (in the dashed box) was magnified 10 times compared to the corresponding region of 1.0–4.5 ppm for the purpose of clarity. Local amplified regions of AXP peaks (overlaid ATP/ADP) for the Ctrl, Iod10 and Iod30 groups (**B**), the Ctrl, Iod10 and D-4F + Iod10 groups (**C**), and the Ctrl, Iod30 and D-4F + Iod30 groups (**D**).

**Figure 3 molecules-26-05123-f003:**
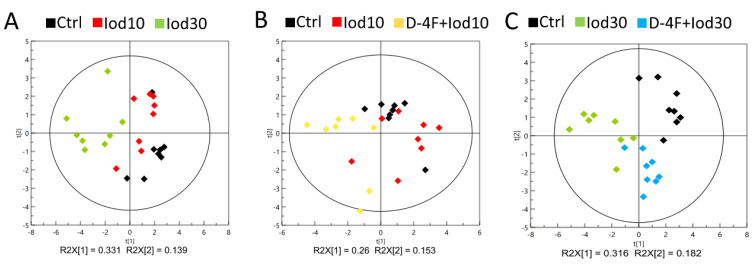
PCA scores plot for ^1^H-NMR spectra recorded on aqueous extracts of HUVECs. (**A**) The Ctrl, Iod10 and Iod30 groups; (**B**) the Ctrl, Iod10 and D-4F + Iod10 groups; and (**C**) the Ctrl, Iod30 and D-4F + Iod30 groups are shown.

**Figure 4 molecules-26-05123-f004:**
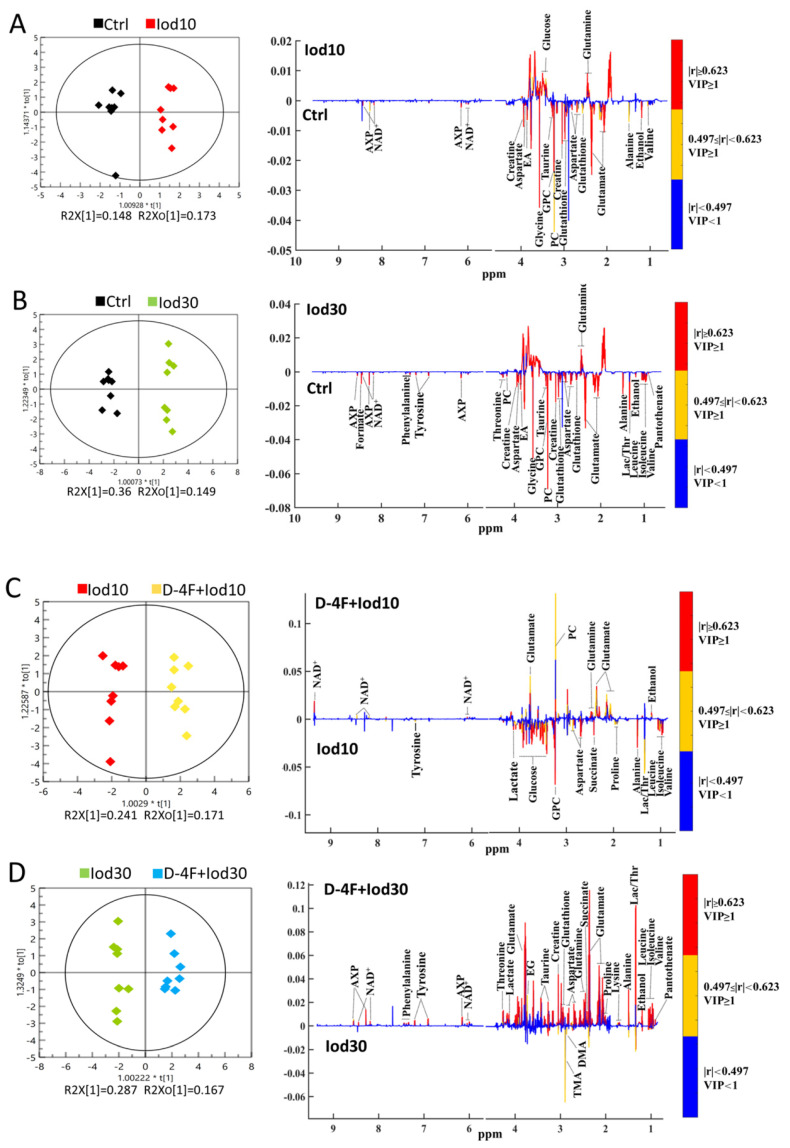
OPLS-DA score plots and corresponding correlation coefficient-coded loading plots for ^1^H-NMR spectra recorded on aqueous extracts of HUVECs. The color pattern of the loading plot is used to identify the significant variables in the class separation. Red, orange and blue denote that the variables are very significant, significant and insignificant, respectively. (**A**) Iod10 vs. Ctrl; (**B**) Iod30 vs. Ctrl; (**C**) D-4F + Iod10 vs. Iod10; and (**D**) D-4F + Iod30 vs. Iod30 were shown.

**Figure 5 molecules-26-05123-f005:**
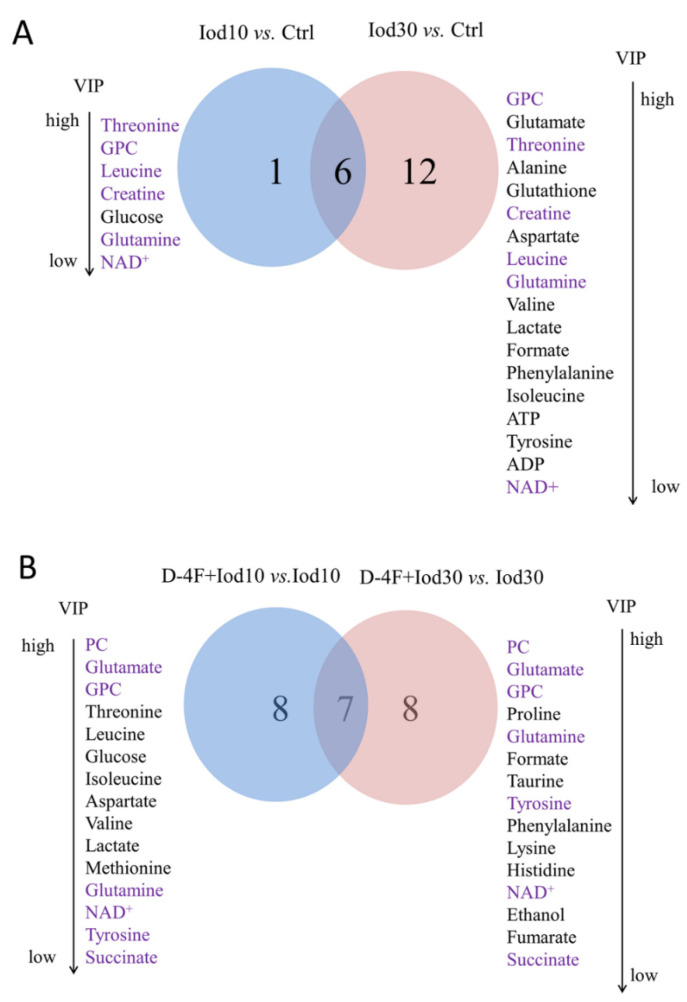
Venn diagrams of characteristic metabolites identified from pair-wise comparisons. (**A**) Iod10 vs. Ctrl and Iod30 vs. Ctrl; and (**B**) D-4F + Iod10 vs. Iod10 and D-4F + Iod30 vs. Iod30 were shown. The characteristic metabolites were identified with two criteria: VIP ≥ 1 calculated from the OPS-DA models; and *p* < 0.05 calculated from the pair-wise *t* tests.

**Figure 6 molecules-26-05123-f006:**
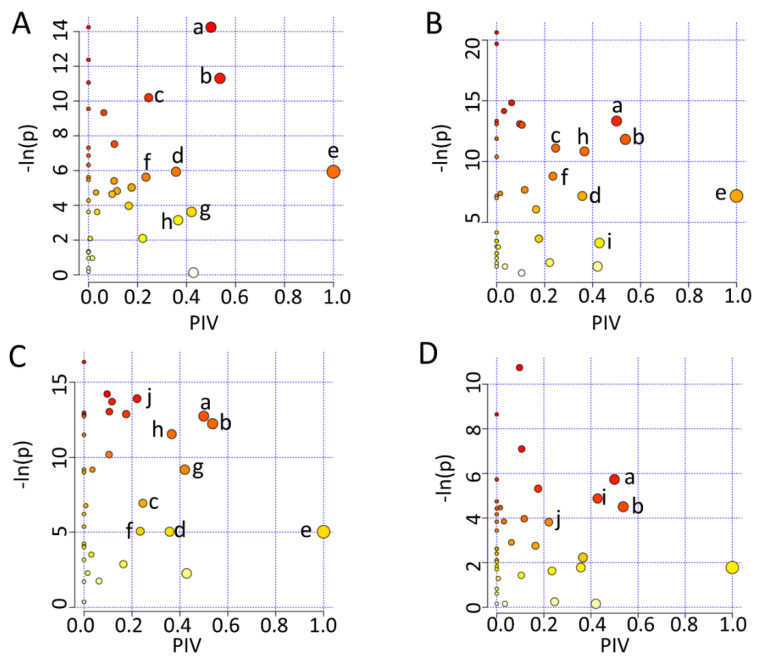
Metabolic pathway analysis. (**A**) Iod10 vs. Ctrl; (**B**) Iod30 vs. Ctrl; (**C**) D-4F + Iod10 vs. Iod10; and (**D**) D-4F + Iod30 vs. Iod30 are shown. Significantly altered metabolic pathways were identified using a combination of metabolite sets enrichment analysis (*p* < 0.05) and topological pathway analysis (PIV > 0.2) based on the absolute concentrations of metabolites. a: glutamine and glutamate metabolism; b: alanine, aspartate and glutamate metabolism; c: glycine, serine and threonine metabolism; d: phenylalanine metabolism; e: phenylalanine, tyrosine and tryptophan biosynthesis; f: nicotinate and nicotinamide metabolism; g: starch and sucrose metabolism; h: glutathione metabolism; i: taurine and hypotaurine metabolism; and j: histidine metabolism.

**Table 1 molecules-26-05123-t001:** Pair-wise comparisons of the metabolite levels between the five groups of HUVECs using Student’s *t* test.

Metabolite	Mean ± Standard Error					Student’s *t* Test			
	Ctrl	Iod10	D-4F + Iod10	Iod30	D-4F + Iod30	Iod10	D-4F + Iod10	Iod30	D-4F + Iod30
						vs. Ctrl	vs. Iod10	vs. Ctrl	vs. Iod30
**Amino acid metabolism**									
Pantothenate	0.13 ± 0.01	0.13 ± 0.01	0.13 ± 0.01	0.11 ± 0.00	0.13 ± 0.01	ns	ns	ns	ns
Leucine	1.35 ± 0.02	1.43 ± 0.02	1.27 ± 0.03	1.26 ± 0.00	1.28 ± 0.02	*	***	***	ns
Isoleucine	0.60 ± 0.01	0.62 ± 0.01	0.55 ± 0.02	0.55 ± 0.00	0.56 ± 0.01	*	***	***	ns
Valine	0.56 ± 0.01	0.57 ± 0.00	0.52 ± 0.01	0.51 ± 0.00	0.51 ± 0.01	ns	***	***	ns
Threonine	4.29 ± 0.07	4.76 ± 0.07	4.41 ± 0.07	4.63 ± 0.11	4.63 ± 0.08	***	**	*	ns
Aspartate	0.85 ± 0.01	0.86 ± 0.02	0.79 ± 0.02	0.76 ± 0.02	0.77 ± 0.01	ns	*	***	ns
Alanine	1.57 ± 0.04	1.53 ± 0.07	1.43 ± 0.02	1.38 ± 0.01	1.34 ± 0.04	ns	ns	***	ns
Lysine	0.14 ± 0.00	0.14 ± 0.01	0.15 ± 0.01	0.14 ± 0.00	0.15 ± 0.00	ns	ns	ns	*
Proline	0.58 ± 0.01	0.58 ± 0.01	0.62 ± 0.01	0.55 ± 0.01	0.60 ± 0.01	ns	**	**	****
Phenylalanine	0.32 ± 0.00	0.33 ± 0.01	0.32 ± 0.01	0.29 ± 0.00	0.30 ± 0.01	ns	*	***	*
Tryosine	0.36 ± 0.00	0.37 ± 0.00	0.36 ± 0.00	0.33 ± 0.00	0.35 ± 0.01	**	*	***	*
Fumarate	0.04 ± 0.00	0.04 ± 0.00	0.04 ± 0.00	0.03 ± 0.00	0.03 ± 0.00	ns	ns	**	*
Succinate	0.01 ± 0.00	0.01 ± 0.00	0.01 ± 0.00	0.02 ± 0.00	0.02 ± 0.00	***	**	***	*
Formate	0.20 ± 0.04	0.15 ± 0.02	0.11 ± 0.01	0.12 ± 0.01	0.09 ± 0.01	ns	ns	ns	*
Histidine	0.07 ± 0.00	0.06 ± 0.00	0.07 ± 0.00	0.07 ± 0.00	0.06 ± 0.00	ns	ns	ns	*
Taurine	0.36 ± 0.01	0.36 ± 0.01	0.33 ± 0.01	0.37 ± 0.01	0.33 ± 0.01	ns	ns	ns	**
Creatine	0.96 ± 0.01	0.89 ± 0.02	0.94 ± 0.02	0.84 ± 0.01	0.86 ± 0.01	**	ns	***	ns
**Glutathione metabolism**									
Glutathione	1.81 ± 0.02	1.75 ± 0.06	1.78 ± 0.03	1.63 ± 0.03	1.64 ± 0.04	ns	ns	***	ns
Glutamate	8.22 ± 0.04	7.89 ± 0.10	8.69 ± 0.08	7.66 ± 0.08	8.08 ± 0.04	**	****	***	***
Glutamine	0.38 ± 0.01	0.43 ± 0.01	0.47 ± 0.01	0.45 ± 0.01	0.49 ± 0.01	***	**	***	**
Methionine	0.39 ± 0.01	0.43 ± 0.01	0.38 ± 0.00	0.36 ± 0.01	0.36 ± 0.01	**	****	*	ns
Glycine	0.10 ± 0.00	0.11 ± 0.00	0.11 ± 0.00	0.12 ± 0.00	0.12 ± 0.00	ns	ns	**	ns
**Glucose metabolism**									
Glucose	0.31 ± 0.03	0.39 ± 0.02	0.23 ± 0.02	0.34 ± 0.03	0.35 ± 0.02	*	****	ns	ns
Lactate	0.36 ± 0.01	0.38 ± 0.01	0.32 ± 0.02	0.42 ± 0.02	0.40 ± 0.01	ns	*	*	ns
**Glycerophospholipid metabolism**									
Choline	0.13 ± 0.01	0.14 ± 0.01	0.14 ± 0.01	0.14 ± 0.01	0.14 ± 0.01	ns	ns	ns	ns
PC	5.19 ± 0.14	5.06 ± 0.05	6.06 ± 0.16	4.97 ± 0.11	5.50 ± 0.08	ns	****	ns	**
GPC	1.45 ± 0.06	1.19 ± 0.04	0.87 ± 0.03	0.92 ± 0.02	0.76 ± 0.02	**	****	***	****
EA	0.08 ± 0.00	0.08 ± 0.00	0.08 ± 0.00	0.08 ± 0.00	0.09 ± 0.00	ns	ns	ns	*
**Others**									
ATP	0.30 ± 0.01	0.30 ± 0.01	0.28 ± 0.02	0.25 ± 0.01	0.27 ± 0.01	ns	ns	*	ns
ADP	0.17 ± 0.01	0.16 ± 0.00	0.16 ± 0.01	0.15 ± 0.01	0.16 ± 0.00	ns	ns	**	ns
NAD+	0.09 ± 0.00	0.09 ± 0.00	0.10 ± 0.00	0.08 ± 0.00	0.09 ± 0.00	***	***	***	**
NADP+	0.07 ± 0.00	0.07 ± 0.00	0.07 ± 0.00	0.06 ± 0.00	0.06 ± 0.00	ns	ns	***	ns

Note: * *p* < 0.05; ** *p* < 0.01; *** *p* < 0.001; **** *p* < 0.0001; Red/Blue stars denote increased/decreased metabolite levels.

## Data Availability

The data that support the findings of this study are available from the corresponding authors upon reasonable request.

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
