# Peer review of "NMR-Based Metabolomic Analysis on the Protective Effects of Apolipoprotein A-I Mimetic Peptide against Contrast Media-Induced Endothelial Dysfunction"

_molecules, 2021, doi:10.3390/molecules26175123_

Round 1

Reviewer 1 Report

In this manuscript, the authors evaluated iodixanol treatment in human umbilical vein endothelial cells (HUVECs) and performed nuclear magnetic resonance (NMR)-based metabolomics related to energy-producing pathways and oxidative stress in HUVEC.

In this manuscript, the authors evaluated iodixanol treatment in human umbilical vein endothelial cells (HUVECs) and performed nuclear magnetic resonance (NMR)-based metabolomics related to energy-producing pathways and oxidative stress in HUVEC.

The manuscript presents interesting results and analyzed using modern techniques. The results are clearly presented. And presents a logical sequence in the presentation of results and consistent with the discussion.

Figure 7 helps readers from all areas to understand the results obtained by the authors.

Reviewer 2 Report

The authors submitted a research paper related to NMR-based metabolomic analysis on the protective effects of 2 apolipoprotein A-I mimetic peptide against contrast media-induced endothelial dysfunction.

Given the comprehensive analysis of investigated problem provided in the introduction section, the hypotheses, as well as main goals of this investigation have been adequately postulated. Methodological approach was carefully chosen and effectively applied, thus providing original findings of high quality. The obtained results have been extensively discussed and scientifically paralleled with findings available so far.

All together, the authors provided new and valuable results leading to better understanding of contrast media-induced endothelial impairments and the protective effects of D-4F for improving endothelial cell dysfunction. Clinical relevance of the submitted paper was very clear throughout the whole manuscript.

Reviewer 3 Report

The paper is interesting and I believe should be published. The metabolomic approach, although complex , was used to study the inhbiting effect of the D-4F peptide on the effects of the contrast agent iodixanol.

  1. the abstract 23-43 appears to some extent ripetitive and it is not comple\\\tely clearly written as the rest of the paper.
  2. A brief description of the mechanism of action of this CM should be given to the reader . Also if the metabolic effects are not completely clear as ìcorrectly written. A brief similarità / dissimilarity with other CM should help the reader to enter in the problem.
  3. The same should be given for the mechanism of action of the D-4F peptide.
  4. The results in Fig. 1 are very difficult to follow whilst the plot at the right side are very effective. Moreover the presentation by numbers of the results of Table 1 may be reported in more visible way ( bars?) . The student tests should be shifted in another table in a more clear way.
  5. The very interesting results of Fig. 2 A are reported in a small format. In several places the term ns is indicated and now at row 263 is finally defined as insignificant, I think that the term to be used is “ poorly significant” .
  6. In Table 1 tryosine should be converted in Tyrosine
  7. The interesting report of Fig. 5 is not clearly explaind in the legend regard to the overlapping areas.
  8. Also the meaning of dots in plot of Fig6 A,B,C and D are really poorly explained in the legend.

Reviewer 4 Report

The authors employ metabolic profiling by NMR spectroscopy to look at the physiological alterations caused by a iodixanol, a radiocontrast agent and the potentially alleviating effected of the D-4F peptide.

The methodology is mostly sound and well described and I think the paper can be recommended for publication once my comments below are addressed.

(1) Remove the pathway analysis - The authors detect and quantify 36 metabolites, the metabolome contains tens of thousands of metabolites, the coverage is way too low to carry out sensible pathway analysis here.

(2) Report effect sizes - In Table 1 the raw data and the significance (p values of a t-test) are reported. Looking at the data, I can see that for most metabolites, changes are very modest. Therefore, I would suggest to include effect sizes (e.g. percentage change) as well as p values to guide the reader.

(3) Explore different normalisation options - The total area normalisation the authors used produces an artefact under certain conditions. If, for example, treatment with Iodixanol leads to higher levels of glucose and glutamine inside the cell and leaves the other compounds unchanged, a total area normalisation would make it appear as if glucose and glutamine were increased while most other compounds were decreased. I think this might have happened here. Maybe the authors can normalise to cell number and re-run their analysis.

(4) A media sample / spetrum to compare the changes in the medium would have been beneficial, but probably too late now.
